# Toward Balanced Continual Learning via Fine-Grained Neuronal Intervention Inspired by Memory Consolidation

## Abstract

Continual learning confronts the fundamental *stability-plasticity dilemma* between preserving previously acquired knowledge and adapting to novel tasks. Existing approaches employ coarse-grained network-level regularization that fails to capture the fine-grained neuronal dynamics essential for effective stability-plasticity orchestration. The human brain resolves this challenge through *memory reconsolidation*—a neural mechanism that selectively reactivates task-relevant memory traces during retrieval, temporarily destabilizing them to enable integration of new information while preserving task-irrelevant memories. Inspired by this neurobiological principle, we introduce K-Recon, a neuron-level continual learning architecture that operationalizes memory reconsolidation through fine-grained neural pathway modulation. Our approach orchestrates stability and plasticity via two complementary components: **(i)** a *Selective Reactivation Module* that performs controlled reactivation and consolidation blockade of task-relevant neuronal clusters and memory amalgamation, and **(ii)** an *Adaptive Consolidation Module* that enforces parameter protection for inactive neuronal clusters while strategically releasing connections from obsolete tasks. This neuron-level intervention is theoretically grounded within a unified optimization framework, enabling seamless integration into existing continual learning paradigms as a plug-and-play enhancement. Extensive evaluation across diverse continual learning benchmarks validates K-Recon's effectiveness as a model-agnostic architectural enhancement. Notably, on CIFAR-100 sequential classification tasks, our framework achieves a remarkable **6.43%** improvement in average incremental accuracy relative to EWC, establishing neuron-level memory reconsolidation as an effective technique for continual learning. Code for experiments is available at https://anonymous.4open.science/r/K_Recon11-CF57

## 1 Introduction

Continual learning aims to enable neural networks to acquire new knowledge while retaining previously learned information, yet remains fundamentally constrained by the *stability-plasticity dilemma* Chaudhry et al. (2019). While stability preserves acquired knowledge, plasticity facilitates adaptation to new tasks—striking an optimal balance between these competing objectives represents one of the most challenging problems in machine learning Wang et al. (2024a). However, excessive emphasis on preserving prior knowledge often impedes effective acquisition of new knowledge, creating a tension that existing methods struggle to resolve. Current approaches primarily address this dilemma at the overall network level Lan et al. (2025), neglecting fine-grained analysis and control at the neuronal level. This results in an inability to effectively identify and dynamically regulate the behavior of active versus inactive neurons, ultimately limiting the model's generalization ability in complex and dynamic environments.

The human brain can adjust the activity of synapses through a memory reconsolidation mechanism when learning new knowledge, dynamically updating old knowledge to adapt to the learning of new knowledge and reinforcing stable old knowledge. During the learning of new knowledge, the memory reconsolidation mechanism consolidates and blocks the synapses of similar memories, reactivating them to an active state for integration, while the synapses of inactive memories that are less

affected are consolidated, forming long-term memories Ritter et al. (2018); Audrain & McAndrews (2022); Gilboa & Marlatte (2017). This process allows for the dynamic updating of old knowledge, providing a paradigm for addressing the plasticity and stability issues at the neuronal level with fine granularity. Consequently, this paper simulates the integration and consolidation processes reflected in memory reconsolidation, activating neurons by identifying the similarity between new and old knowledge and seeking Pareto optimal solutions to achieve effective knowledge integration, protecting and consolidating the less affected inactive synapses, and reactivating redundant synapses, thereby reinforcing important knowledge and fully unleashing the model's learning capacity for new tasks.

The study begins with an in-depth theoretical analysis of traditional continual learning methods, unifying them under a regularization framework. This framework introduces constraints in the loss function to limit changes in model parameters, thereby partially mitigating catastrophic forgetting. Building on this unified perspective, we design memory integration and consolidation modules inspired by the reconsolidation mechanism. Specifically, these modules automatically adjust the integration ratio between old and new task representations based on their similarity and importance, selectively suppressing conflicting information and mitigating forgetting. This enhances the efficiency of new memory formation and improves representational consistency between old and new memories. Furthermore, our approach is designed to be plug-and-play, allowing seamless integration into various existing continual learning frameworks without significant modifications to the original model architecture. To validate the effectiveness of the proposed theory and method, we conducted evaluations on three standard benchmark datasets for continual learning. Experimental results demonstrate that the K-RECON method effectively enhances the performance of various continual learning approaches. Additionally, neuron-level analysis provides evidence of our method's efficacy in integrating old and new knowledge.

In summary, our contributions can be highlighted as follows:

- Drawing inspiration from the biological learning paradigm, we propose a continual learning method based on the memory reconsolidation mechanism called K-RECON.
- From the perspective of cognitive science and generalization, the mechanism of each module is deeply analyzed, and the reasons for its good performance are explained.
- We conduct extensive experimental verification on multiple continuous learning benchmark tasks, and conduct comprehensive ablation experiments to analyze the contribution of each module.

## 2 REVISITING CL METHODS FROM A REGULARIZATION PERSPECTIVE

This work first unifies various continual learning methods from the perspective of applying regularization to parameter updates, and then conducts neuron-level analysis by simulating the memory reconsolidation mechanism in the learning process of the human brain through this unified perspective. From this angle, all mainstream continual learning methods can be viewed as applying secondary regularization to parameter updates.

$$\mathcal{L}(\theta) = \mathcal{L}_{\text{new}}(\theta) + \frac{1}{2}(\theta - \theta^*)^\top \Omega (\theta - \theta^*), \tag{1}$$

where $\theta^*$ denotes the optimal parameters for previous tasks, and $\Omega \in \mathbb{R}^{P \times P}$ is a diagonal or general quadratic-form matrix encoding importance weights for each parameter dimension or direction. In this section, we use this unified framework to discuss how parameter regularization, replay, distillation, and parameter isolation correspond to different choices of $\Omega$.

**Parameter Regularization** This approach explicitly constructs a diagonal matrix $\Omega = \lambda \operatorname{diag}(F)$, where $F_i$ is the importance score (e.g., Fisher information) of parameter $i$ on prior tasks. The optimization objective is:

$$\mathcal{L}_{\text{reg}}(\theta) = \mathcal{L}(\theta; D_{\text{new}}) + \frac{\lambda}{2} \sum_i F_i (\theta_i - \theta_i^*)^2. \tag{2}$$

Here, $\Omega$ is statically estimated a priori and does not change during new-task training.

**Replay Strategy**  Replay methods implicitly form a matrix $\Omega$ by computing the loss over old data $D_{\text{replay}}$. Concretely, optimizing

$$\mathcal{L}_{\text{replay}} = \mathcal{L}(\theta; D_{\text{new}}) + \beta\,\mathcal{L}(\theta; D_{\text{replay}}) \tag{3}$$

is equivalent to using a diagonal $\Omega$ with entries

$$\Omega_{ii} = \beta\,\mathbb{E}_{x \sim D_{\text{replay}}}\big[\partial_{\theta_i}\ell(\theta^*, x)^2\big], \tag{4}$$

where $\ell$ is the per-sample loss. This $\Omega$ can be viewed as a weighted estimate of the squared empirical gradients and is updated dynamically during training.

**Knowledge Distillation**  Distillation methods impose a KL-divergence regularization on old inputs, resulting in

$$\Omega \;=\; \mathbb{E}_x[J_x^\top W_x J_x], \tag{5}$$

and the loss is:

$$\begin{aligned}
\mathcal{L}_{\text{KD}} &= \mathbb{E}_{x \sim D_{\text{replay}}}\big[\text{KL}(f_{\theta^*}(x)\,\|\,f_\theta(x))\big] \\
&\approx \tfrac{1}{2}(\theta - \theta^*)^\top \Omega (\theta - \theta^*),
\end{aligned} \tag{6}$$

This $\Omega$ is also refined as new samples are processed, and $W_x = \nabla_f^2\big[\text{KL}(f_{\theta^*}(x)\,\|\,f)\big]_{f=f_{\theta^*}(x)}$, and $J_x = \partial_\theta f_\theta(x)\big|_{\theta^*}$

**Parameter Isolation**  Freezing a subset of parameters via a binary mask $m \in \{0,1\}^P$ can be interpreted as a setting

$$\Omega_{ii} = \begin{cases} 0, & m_i = 1, \\ +\infty, & m_i = 0, \end{cases} \quad \mathcal{L}_{\text{iso}}(\theta) = \tfrac{1}{2}(\theta - \theta^*)^\top \Omega\,(\theta - \theta^*). \tag{7}$$

Here, $\Omega$ is both static and binary: Unfrozen dimensions incur no penalty, while frozen dimensions face a hard constraint.

**Summary**  Within the unified framework, different methods only adopt different forms of regularization matrices, but their core lies in the quadratic regularizer. In the next section, based on this unified framework, we will introduce a neuron-level memory reconsolidation mechanism to simulate the processes of selective reactivation and adaptive consolidation of neuronal clusters, thereby achieving a more flexible trade-off between stability and plasticity.

## 3 METHOD

To address the stability-plasticity dilemma in continual learning, we propose our main contribution: K-RECON. This is a neuron-level plug-and-play method designed to simulate the processes of selective reactivation of neuronal clusters and adaptive consolidation inherent in the brain's memory reconsolidation mechanism. It constructs two complementary components to achieve the following during new task learning: selectively activating task-relevant neuronal clusters for consolidation blocking and memory amalgamation, adaptively consolidating persistent memories, and releasing connection constraints from obsolete tasks—ultimately orchestrating stability and plasticity throughout the task learning process.

### 3.1 SELECTIVE REACTIVATION MODULE

We simulate the selective reactivation of neuronal clusters associated with old tasks, as well as the integration process of knowledge from old and new tasks, which occurs in the human brain during the learning of new and existing knowledge. To facilitate efficient learning of new tasks, the human brain leverages the function of consolidation suppression to prevent information from old tasks that are highly similar to the new task from being consolidated into long-term memory. Instead, it reactivates the neuronal clusters of these old tasks and performs information integration to maximize the overall benefit between old and new tasks.

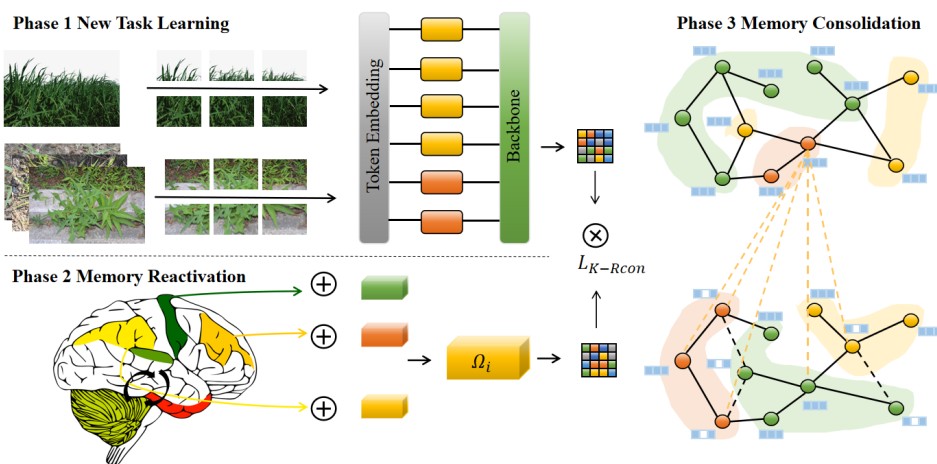

Figure 1: Overall model architecture. It is designed to simulate the processes of selective memory reactivation and adaptive consolidation in the memory reconsolidation mechanism. When learning a new task, the model operates as follows: (1) It acquires indicators such as the relevance and importance of neurons associated with the new task. (2) It initiates a consolidation blocking process, selectively reactivating neuronal clusters that exhibit high similarity to the new task and restoring their active state—this enables the integration of information related to both old and new tasks. (3) The model protects neuronal clusters that are highly important and less susceptible to the influence of the new task, consolidating them into long-term memory. During this process, it also strategically lifts connection constraints from obsolete tasks to enhance the model's plasticity, ultimately achieving a balance between plasticity and stability.

To simulate this process, we adopt a similarity-based Pareto optimization approach to model the interrelationships between neuronal clusters across tasks. Specifically, we calculate similarity to identify the segments suitable for integration, and utilize Pareto optimization to find the optimal trade-off among multiple tasks—ultimately achieving knowledge integration between old and new tasks.

During task training, the importance of parameters reflects the degree to which different tasks depend on specific model parameters. Notably, tasks with large differences in knowledge distribution often exhibit significantly distinct parameter importance patterns. Therefore, we first estimate the importance of each task (denoted as $\Omega^{(1)}, \Omega^{(2)}, \ldots, \Omega^{(t)}$) to construct task information indicators. Subsequently, we calculate the cosine similarity between the importance matrix $\Omega^{(i)}$ of each historical task and the importance matrix $\Omega^{(t)}$ of the current task.

$$\text{sim}(i) = \frac{\Omega^{(i)} \cdot \Omega^{(t)}}{\|\Omega^{(i)}\| \cdot \|\Omega^{(t)}\| + \epsilon} \tag{8}$$

The $\epsilon$ is used to prevent division by zero. Then the weights are preliminarily fused by this similarity normalization result.

$$\omega_i = \frac{\max(0, \text{sim}(i))}{\sum_j \max(0, \text{sim}(j)) + \epsilon} \tag{9}$$

$$\Omega_{agg} = \sum_i \omega_i \cdot \Omega^{(i)}. \tag{10}$$

Although cosine similarity can align the relationship between new and old learning tasks, it cannot fully eliminate the interference between them. To address this issue, we introduce a Pareto optimality mechanism that minimizes the deviation between the aggregated importance vector $\Omega_{agg}$ and each historical importance vector $\Omega^{(k)}$. The core rationale behind this design lies in the fact that Pareto

optimization comprehensively takes into account the importance of each task, thereby achieving an optimal balance between old and new tasks.

$$\Omega_i = \min_{\omega_{agg}} \sum_{k=1}^{t} \lambda_k ||\Omega_{agg} - \Omega^{(k)}||. \tag{11}$$

Where $\lambda_k$ is the adaptive weight used to minimize the conflict between the final task knowledge and the previous tasks, resulting in the loss function of cosine similarity based Pareto optimization.

$$L = L_{new} + \frac{\lambda}{2}\Omega_i(\theta_i - \theta_i^*)^2. \tag{12}$$

### 3.2 ADAPTIVE CONSOLIDATION MODULE

During the memory reconsolidation process, synapses unaffected by new knowledge undergo consolidation to form stable long-term memories, while redundant information from obsolete tasks is actively forgotten. This process not only helps the brain consolidate core knowledge but also promotes the active forgetting of invalid or obsolete information, thereby facilitating the efficient adaptation to new tasks. To further simulate the adaptive reconsolidation process in the memory reconsolidation mechanism, we apply attention-based regularization constraints at multiple levels. These constraints serve two core functions: (1) Protect stable neuronal clusters that are not affected by new knowledge, enabling them to form long-term memories and enhancing stability; (2) Release connection constraints from invalid or obsolete tasks to improve the model's plasticity.

$$a_i = \frac{exp(f \cdot \Omega_i)}{\sum_j exp(f \cdot \Omega_i)}. \tag{13}$$

$$\widetilde{\Omega}_i = a_i \cdot \Omega_i. \tag{14}$$

Here, $f$ is the attention amplification factor used to differentiate the importance of various task information, and $\tilde{\omega}_i$ represents the importance after attention amplification. By leveraging the selective forgetting characteristic of memory integration, we simulate the adaptive consolidation process by releasing neurons with importance below the threshold $\varphi_i$ and protecting stable neurons with importance above the threshold $\varphi_j$. This enables parameter protection for inactive neuronal clusters while strategically lifting connection constraints from obsolete tasks.

## 4 THEORETICAL ANALYSIS

**NTK Linearization and Closed-Form Update** Under the infinite-width NTK approximation (with the Jacobian held constant), taking the current parameters $\theta_{t-1}$ as the expansion point, we perform a first-order Taylor expansion of the network output on task $t$'s inputs $X_t$:

$$f_\theta(X_t) \approx f_{\theta_{t-1}}(X_t) + J_t(\theta - \theta_{t-1}), \tag{15}$$

where $J_t = J_{\theta_{t-1}}(X_t) \in \mathbb{R}^{n_t \times P}$. Let the residual be $e_t = f_{\theta_{t-1}}(X_t) - y_t$, and add the NC-ParetoCL regularizer to the mean-squared loss:

$$\mathcal{L}_t(\theta) = \frac{1}{n_t}\left\|J_t(\theta - \theta_{t-1}) - e_t\right\|_2^2 + \frac{\lambda}{2}(\theta - \theta_{t-1})^\top\Omega_t(\theta - \theta_{t-1}). \tag{16}$$

Setting the gradient to zero yields the parameter update in closed form:

$$\Delta\theta_t = \theta_t - \theta_{t-1} = -\left(J_t^\top J_t + \lambda\,\Omega_t\right)^{-1}J_t^\top e_t. \tag{17}$$

This solution clearly shows that $\Omega_t$ acts as an additional quadratic regularizer in parameter space, enhancing smoothness in directions where gradients overlap (*i.e.*, the subspace spanned by $J_t$), thereby suppressing large parameter shifts. Designing $\Omega_t$ focuses on balancing "stability" and "plasticity"—too large impedes learning of new tasks, too small fails to mitigate forgetting.

**A General Upper Bound on Forgetting**  In the infinite-width NTK regime we can concretely quantify how much training on task $t$ perturbs the outputs on any earlier task $s$. The following result shows that the extent of forgetting is governed by the interplay between the inter-task kernel overlap $K_{st}$ and the spectral strength of the regularized kernel $K_{tt} + \lambda \widetilde{K}_t$.

**Theorem 4.1** (Function-Space Forgetting Upper Bound). *Under the infinite-width NTK approximation, define*

$$\Delta f_s \;=\; f_{\theta_t}(X_s) - f_{\theta_{t-1}}(X_s) \;=\; - K_{st} \left( K_{tt} + \lambda \, \widetilde{K}_t \right)^{-1} e_t, \tag{18}$$

*where $K_{st} = J_s J_t^\top$, $K_{tt} = J_t J_t^\top$, $\widetilde{K}_t = J_t \, \Omega_t \, J_t^\top$ and $e_t = f_{\theta_{t-1}}(X_t) - y_t$. Then for any earlier task $s < t$, the spectral norm of its output change satisfies*

$$\|\Delta f_s\|_2 \leq \frac{\|K_{st}\|_2}{\sigma_{\min}\!\left( K_{tt} + \lambda \, \widetilde{K}_t \right)} \, \|e_t\|_2. \tag{19}$$

**Remark 4.1.**  *(Proof in Appendix.) This theorem shows that reducing the inter-task kernel overlap $\|K_{st}\|_2$ or increasing the minimum eigenvalue of the regularized kernel matrix $K_{tt} + \lambda \widetilde{K}_t$ both directly tighten the upper bound on catastrophic forgetting, offering a clear spectral-view guideline for designing $\Omega_t$.*

**Forgetting Suppression**  Building on this general bound, we now introduce an explicit spectral condition on the memory matrix $\Omega_t$ that yields a tighter, task-suppression guarantee.

**Theorem 4.2** (Forgetting Suppression). *Assume the memory matrix $\Omega_t$ satisfies $\sigma_{\min}(\Omega_t) \geq \mu > 0$ on the gradient-overlap subspace $\mathrm{span}\{J_t^\top J_s : s < t\}$. Then for any earlier task $s < t$, the following forgetting bound holds:*

$$\|\Delta f_s\|_2 \leq \frac{\|K_{st}\|_2}{\sigma_{\min}(K_{tt}) \, (1 + \lambda \, \mu)} \, \|e_t\|_2. \tag{20}$$

**Remark 4.2.**  *(Proof in Appendix.) This proposition explicitly introduces the spectral lower bound $\mu$ of $\Omega_t$ into the denominator of the forgetting bound—larger $\mu$ yields a larger denominator and thus a tighter bound, enhancing forgetting suppression. Hence, constructing $\Omega_t$ with sufficient spectral energy on the overlap subspace is crucial.*

**Enhancing $\mu$ via K-RECON**  Let the historical task importance matrices be $\{\Omega^{(k)}\}_{k=1}^t$, weighted by cosine similarity to the current task:

$$\Omega_t = \sum_{k=1}^{t} \omega_{t,k} \, \Omega^{(k)}, \quad \omega_{t,k} \propto \cos\!\left(\Omega^{(k)}, \Omega^{(t)}\right), \tag{21}$$

and perform Pareto minimization during integration to maximize $\sigma_{\min}(\Omega_t)$ while minimizing inter-task conflict. This strategy preserves directions most relevant to the current task, automatically reinforcing spectral energy in the overlap subspace to maximize $\mu$ without compromising new-task performance. Finally, we threshold low-importance eigenvalues of $\Omega_t$ (below $\varphi$) to zero, retaining spectral energy only in high-overlap modes:

$$\Omega_t \leftarrow U \, \mathrm{diag}(\max(\sigma_i, \varphi)) \, U^\top, \tag{22}$$

where $\{\sigma_i\}$ are the eigenvalues of $\Omega_t$ and $U$ the eigenvector matrix. This operation frees directions orthogonal to the current task for plasticity, while preserving $\mu$ in the overlap subspace, achieving a balance of "stability + adaptability." This conclusion theoretically proves the feasibility of integrating new and old tasks using the reconsolidation principle in cognitive theory.

## 5 EXPERIMENTS

### 5.1 EXPERIMENTAL SETUP

We conduct experiments on classical continual learning datasets such as CIFAR-10, CIFAR-100 Krizhevsky & Hinton (2009), and Tiny-ImageNet Abai & Rajmalwar (2019) to evaluate the

Table 1: Task-IL and Class-IL overall accuracy on CIFAR-10, CIFAR-100, and Tiny-ImageNet with memory size 500. '——' indicates not applicable.

| Method | CIFAR-10 | | CIFAR-100 | | Tiny-ImageNet | |
|---|---|---|---|---|---|---|
| | Class-IL | Task-IL | Class-IL | Task-IL | Class-IL | Task-IL |
| fine-tuning | $19.62_{\pm0.05}$ | $61.02_{\pm3.33}$ | $9.29_{\pm0.33}$ | $33.78_{\pm0.42}$ | $7.92_{\pm0.26}$ | $18.31_{\pm0.68}$ |
| Joint train | $92.20_{\pm0.15}$ | $98.31_{\pm0.12}$ | $71.32_{\pm0.21}$ | $91.31_{\pm0.17}$ | $59.99_{\pm0.19}$ | $82.04_{\pm0.10}$ |
| NCL | $19.61_{\pm0.05}$ | $63.29_{\pm2.35}$ | $9.70_{\pm0.23}$ | $28.07_{\pm1.96}$ | $8.46_{\pm0.22}$ | $15.85_{\pm0.58}$ |
| LwF | $19.53_{\pm0.32}$ | $64.49_{\pm4.06}$ | $8.12_{\pm0.28}$ | $20.92_{\pm2.32}$ | $7.56_{\pm0.36}$ | $16.29_{\pm0.87}$ |
| GPM | —— | $90.68_{\pm3.29}$ | —— | $72.48_{\pm0.40}$ | —— | —— |
| HAT | —— | $92.56_{\pm0.78}$ | —— | $72.06_{\pm0.50}$ | —— | —— |
| A-GEM | $22.67_{\pm0.57}$ | $89.48_{\pm1.45}$ | $9.30_{\pm0.32}$ | $48.06_{\pm0.57}$ | $8.06_{\pm0.04}$ | $25.33_{\pm0.49}$ |
| ER | $57.74_{\pm0.27}$ | $93.61_{\pm0.27}$ | $20.98_{\pm0.35}$ | $73.37_{\pm0.43}$ | $9.99_{\pm0.29}$ | $48.64_{\pm0.46}$ |
| CPR | $19.61_{\pm3.67}$ | $68.23_{\pm3.87}$ | $8.42_{\pm0.37}$ | $21.43_{\pm2.57}$ | $7.67_{\pm0.23}$ | $15.58_{\pm0.91}$ |
| GSS | $49.73_{\pm4.78}$ | $91.02_{\pm1.57}$ | $13.60_{\pm2.98}$ | $57.50_{\pm1.93}$ | —— | —— |
| HAL | $41.79_{\pm4.46}$ | $84.54_{\pm2.36}$ | $9.05_{\pm2.76}$ | $42.94_{\pm1.80}$ | —— | —— |
| SI | $19.48_{\pm0.17}$ | $68.05_{\pm5.91}$ | $\mathbf{9.41}_{\pm0.24}$ | $31.08_{\pm1.65}$ | $6.58_{\pm0.31}$ | $36.32_{\pm0.13}$ |
| +K-RECON | $\mathbf{20.94}_{\pm1.21}$ | $\mathbf{69.32}_{\pm2.71}$ | $8.79_{\pm1.92}$ | $\mathbf{32.19}_{\pm2.27}$ | $\mathbf{7.82}_{\pm0.54}$ | $\mathbf{37.25}_{\pm0.68}$ |
| oEWC | $19.49_{\pm0.12}$ | $64.31_{\pm4.31}$ | $8.24_{\pm0.21}$ | $21.20_{\pm2.08}$ | $7.42_{\pm0.31}$ | $15.19_{\pm0.82}$ |
| +refresh | $20.37_{\pm0.65}$ | $66.89_{\pm2.57}$ | $8.78_{\pm0.42}$ | $23.31_{\pm1.87}$ | $7.83_{\pm0.15}$ | $17.32_{\pm0.85}$ |
| +K-RECON | $\mathbf{20.53}_{\pm0.72}$ | $\mathbf{68.52}_{\pm1.37}$ | $\mathbf{8.94}_{\pm0.82}$ | $\mathbf{27.63}_{\pm1.62}$ | $\mathbf{8.27}_{\pm0.76}$ | $\mathbf{18.28}_{\pm0.32}$ |
| DER++ | $72.70_{\pm1.36}$ | $93.88_{\pm0.50}$ | $36.37_{\pm0.85}$ | $75.76_{\pm0.60}$ | $19.38_{\pm1.41}$ | $51.91_{\pm0.68}$ |
| +refresh | $74.42_{\pm0.82}$ | $\mathbf{94.64}_{\pm0.38}$ | $38.49_{\pm0.76}$ | $77.71_{\pm0.85}$ | $20.81_{\pm1.28}$ | $54.06_{\pm0.79}$ |
| +K-RECON | $\mathbf{75.56}_{\pm0.61}$ | $94.23_{\pm2.15}$ | $\mathbf{38.51}_{\pm0.82}$ | $\mathbf{78.95}_{\pm1.68}$ | $\mathbf{21.03}_{\pm1.54}$ | $\mathbf{55.23}_{\pm1.24}$ |

effectiveness of our proposed method in various continual learning scenarios. Specifically, we assess the method on Task-Incremental Learning (Task-IL) and Class-Incremental Learning (Class-IL). Following the experimental setup in Buzzega et al. (2020), the CIFAR-10, CIFAR-100, and Tiny-ImageNet datasets are divided accordingly. Based on the fact that each task contains two non-overlapping classes, CIFAR-10 is split into five different training tasks. CIFAR-100 is divided into ten training tasks, each containing ten distinct classes. Finally, Tiny-ImageNet is divided into ten training tasks, each covering twenty different categories.

## 5.2 BASELINE MODEL

In order to fully verify the general applicability and effectiveness of our proposed plug-and-play method for different continuous learning methods, we widely selected the following aspects of continuous learning methods for verification. (1) Regularized continuous learning method, It includes Synaptic Intelligence (SI) Zenke et al. (2017), Memory Aware Synapses (MAS) Aljundi et al. (2018), and Online Elastic Weight Consolidation (oEWC) Schwarz et al. (2018), Learning without Forgetting (LwF) Li & Hoiem (2016), classifier-projection Regularization (CPR) Cha et al. (2021), Gradient Projection Memory (GPM) Saha et al. (2021), and refresh learning(refresh) Wang et al. (2024b). (2) Replay-based continuous learning methods: ER Chaudhry et al. (2019), GEM Chaudhry et al. (2018), GSS Aljundi et al. (2020), DER++ Buzzega et al. (2020), HAL Chaudhry et al. (2021); (3) Architecture-based continuous learning method: HAT Serra et al. (2018); (4) Bayesian-based Continuous learning method, NCL Kao et al. (2021).

## 5.3 EXPERIMENTAL RESULTS

We present the average accuracy of various continual learning methods on task-incremental learning and class-incremental learning tasks in Table 1. The results demonstrate that our proposed K-RECON strategy consistently improves the performance of different continual learning methods while serving as a plug-and-play module. The performance gains of our approach over the refresh mechanism are particularly notable. It is worth emphasizing that K-RECON significantly enhances the performance of various methods in continual learning scenarios, especially on the

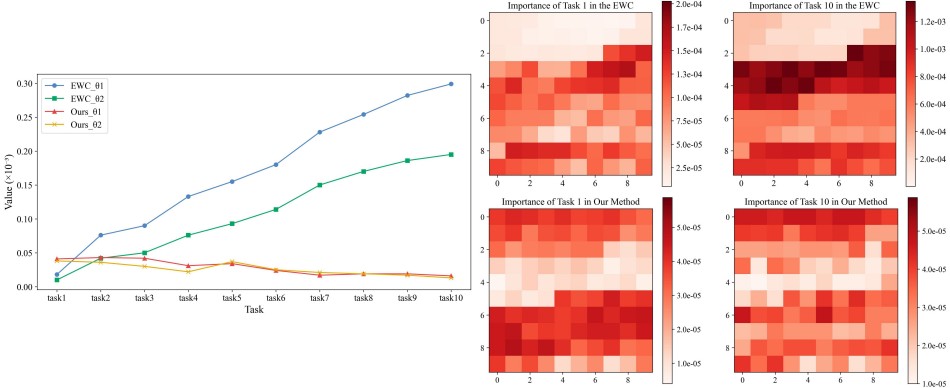

Figure 2: (Left) The changes in the importance of two randomly selected weights demonstrate that the regularization enhancement speed of our method is significantly lower than that of EWC. (Right) The image shows the differences in importance of the top 100 weights between the first training and the last training, indicating that our method can effectively alleviate the problem of excessive model regularization intensity.

Table 2: Table of hyperparameter analysis results

| Method | CIFAR-10 | | CIFAR-100 | | Tiny-ImageNet | |
| | Class-IL | Task-IL | Class-IL | Task-IL | Class-IL | Task-IL |
| --- | --- | --- | --- | --- | --- | --- |
| SI | $19.48_{\pm 0.17}$ | $68.05_{\pm 5.91}$ | $9.41_{\pm 0.24}$ | $31.08_{\pm 1.65}$ | $6.58_{\pm 0.31}$ | $36.32_{\pm 0.13}$ |
| +K-RECON(F=10.0,$\varphi = 0.1$) | $19.94_{\pm 1.21}$ | $68.32_{\pm 2.71}$ | $8.79_{\pm 1.92}$ | $31.19_{\pm 2.27}$ | $6.82_{\pm 0.54}$ | $37.25_{\pm 0.68}$ |
| +K-RECON(F=10.0,$\varphi = 0.3$) | $19.84_{\pm 0.92}$ | $67.98_{\pm 2.03}$ | $8.53_{\pm 2.15}$ | $30.85_{\pm 2.34}$ | $6.51_{\pm 0.54}$ | $36.55_{\pm 0.21}$ |
| +K-RECON(F=8.0,$\varphi = 0.1$) | $18.94_{\pm 1.86}$ | $67.32_{\pm 2.51}$ | $8.15_{\pm 2.31}$ | $30.86_{\pm 1.89}$ | $6.65_{\pm 0.84}$ | $37.35_{\pm 1.53}$ |
| oEWC | $19.49_{\pm 0.12}$ | $64.31_{\pm 4.31}$ | $8.24_{\pm 0.21}$ | $21.2_{\pm 2.08}$ | $7.42_{\pm 0.31}$ | $15.19_{\pm 0.82}$ |
| +refresh | $20.37_{\pm 0.65}$ | $66.89_{\pm 2.57}$ | $8.78_{\pm 0.42}$ | $23.31_{\pm 1.87}$ | $7.83_{\pm 0.15}$ | $17.32_{\pm 0.85}$ |
| +K-RECON(F=10.0,$\varphi = 0.1$) | $20.53_{\pm 0.72}$ | $68.52_{\pm 1.37}$ | $8.94_{\pm 0.82}$ | $29.63_{\pm 1.62}$ | $8.27_{\pm 0.76}$ | $18.28_{\pm 0.32}$ |
| +K-RECON(F=10.0,$\varphi = 0.3$) | $19.65_{\pm 0.25}$ | $68.62_{\pm 1.24}$ | $8.84_{\pm 0.67}$ | $28.93_{\pm 1.45}$ | $8.10_{\pm 0.25}$ | $18.11_{\pm 0.24}$ |
| +K-RECON(F=8.0,$\varphi = 0.1$) | $20.13_{\pm 1.52}$ | $68.31_{\pm 1.51}$ | $8.64_{\pm 0.68}$ | $29.54_{\pm 1.08}$ | $7.99_{\pm 1.58}$ | $17.89_{\pm 1.53}$ |

Tiny-ImageNet and CIFAR-100 datasets, outperforming SI and DER++ respectively. This excellent performance fully demonstrates the effectiveness of our approach in promoting the balance between plasticity and stability at the neuronal level, while also verifying the positive outcomes of combining the memory reconsolidation mechanism with continual learning methods. Due to space constraints, the detailed results regarding model knowledge transfer are provided in the appendix.

## 5.4 ABLATION EXPERIMENTS AND HYPERPARAMETER ANALYSIS

### 5.4.1 ANALYSIS OF THE SELECTIVE REACTIVATION MODULE

By freezing the selective reactivation module, we further analyze the effects of the consolidation suppression and reactivation processes that simulate the reconsolidation mechanism as shown in Figure 2. This analysis verifies that, based on inter-task relationships, the importance of neuronal clusters can be dynamically regulated through a similarity-guided integration approach.

As illustrated in the figure, the neuron cluster selective reactivation module, by simulating the neuron reactivation and consolidation blockade processes in the memory reconsolidation, enables the model to adapt to new task information more efficiently during continual learning and promotes the integration of old and new knowledge. It effectively alleviates the problem of excessively high regularization intensity, thereby reducing the inhibition of model plasticity during training, and also successfully addresses the issue of dependence on the training order in the training process.

### 5.4.2 Hyperparameter analysis

The proposed K-Recon method needs to set the hyperparameters including the quantile threshold for adaptive neuronal consolidation and the attention method factor F in the training process. In order to evaluate the influence of different hyperparameter selection on the experimental results, we carried out several experiments respectively, and the results are shown in Tab. 2.

As can be seen from the table, our method is not sensitive to hyperparameter settings, which also reflects its ease of application and effectiveness as a plug-and-play method. Specifically, under different configurations of the attention amplification factor F and the adaptive consolidation threshold $\varphi$, the experimental results remain stable across various datasets and learning scenarios. This robustness implies that the effectiveness of our method—achieved by simulating the memory reconsolidation mechanism—does not depend on carefully designed hyperparameters.Furthermore, the method consistently maintains stable performance across different tasks, highlighting its practicality and adaptability: it can be seamlessly integrated into a variety of existing continual learning frameworks without complex reconfiguration. This plug-and-play nature not only reduces deployment difficulty but also enhances its generalizability and scalability in diverse continual learning environments.

## 6 Related Work

Continual learning (CL) aims to enable models to learn efficiently from a stream of tasks, allowing them to adapt to new tasks effectively while maintaining performance on previously learned tasks. It also facilitates knowledge transfer and avoids catastrophic forgetting.

**Regularization-based methods** mitigate forgetting by constraining updates on important parameters. Representative techniques include Synaptic Intelligence (SI) Zenke et al. (2017), Memory Aware Synapses (MAS) Aljundi et al. (2018), and Online Elastic Weight Consolidation (oEWC) Schwarz et al. (2018). These methods often rely on static estimates of parameter importance, which can lead to overly rigid learning as tasks accumulate. Recent advances such as the Plasticity Gradient method Kang et al. (2022) introduce adaptive regularization strength to better handle this trade-off.

**Replay-based methods** such as Experience Replay (ER) Chaudhry et al. (2019), Gradient Episodic Memory (GEM) Chaudhry et al. (2018), and Dark Experience Replay++ (DER++) Buzzega et al. (2020) store or regenerate a subset of past data for joint training. Meta-Experience Replay (MER) Riemer et al. (2018) further enhances task relevance modeling through meta-optimization, enabling more effective task alignment. However, memory-based approaches are often constrained by scalability and privacy concerns.

**Architecture-based methods** like Hard Attention to the Task (HAT) Serra et al. (2018) isolate task-specific parameters through masking or gating mechanisms. While these methods minimize interference, they may limit knowledge sharing across tasks and suffer from capacity saturation.

## 7 Conclusion

This paper proposes a unified framework that integrates various existing continual learning methods. Inspired by the memory reconsolidation mechanism in the human brain's learning process, this paper finely simulates the selective reactivation and adaptive consolidation processes of this mechanism at the neuronal level, and innovatively introduces an optimization method called K-Recon. As a plug-and-play solution, K-Recon can be seamlessly integrated into various existing continual learning methods and improve their generalization performance without complex parameter tuning. Extensive experiments on multiple continual learning datasets have verified the effectiveness of the proposed method.

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

# A APPENDIX

## A.1 RECAST EXISTING CL METHODS INTO OUR UNIFIED AND GENERAL FRAMEWORK

Replay methods explicitly store or generate data from previous tasks and optimize:

$$\mathcal{L}_{\text{replay}}(\theta) = \mathcal{L}_{\text{new}}(\theta) + \lambda \mathcal{L}_{\text{old}}(\theta) \tag{23}$$

To relate this to parameter regularization, we approximate $\mathcal{L}_{\text{old}}$ using a second-order Taylor expansion around the previous optimum $\theta^*$:

$$\mathcal{L}_{\text{old}}(\theta) \approx \mathcal{L}_{\text{old}}(\theta^*) + (\theta - \theta^*)^\top \nabla_\theta \mathcal{L}_{\text{old}}(\theta^*) \\ + \frac{1}{2}(\theta - \theta^*)^\top H(\theta - \theta^*) \tag{24}$$

Since $\theta^*$ minimizes $\mathcal{L}_{\text{old}}$, the gradient term vanishes:

$$\nabla_\theta \mathcal{L}_{\text{old}}(\theta^*) = 0 \tag{25}$$

Thus, we obtain:

$$\mathcal{L}_{\text{old}}(\theta) \approx \mathcal{L}_{\text{old}}(\theta^*) + \frac{1}{2}(\theta - \theta^*)^\top H(\theta - \theta^*) \tag{26}$$

Knowledge distillation-based continual learning methods regularize the output distribution to remain close to that of the old model:

$$\min_\theta \ \mathcal{L}_{\text{new}}(\theta) + \lambda \cdot \text{KL}\big(f_\theta(x) \, \| \, f_{\theta^*}(x)\big) \tag{27}$$

Locally linearizing $f_\theta(x)$ around $\theta^*$:

$$f_\theta(x) \approx f_{\theta^*}(x) + J_x(\theta - \theta^*) \tag{28}$$

The KL divergence can then be approximated as:

$$\text{KL}\big(f_\theta(x) \, \| \, f_{\theta^*}(x)\big) \approx C + \frac{1}{2}(\theta - \theta^*)^\top J_x^\top S_x^{-1} J_x(\theta - \theta^*) \tag{29}$$

Parameter isolation methods freeze part of the parameters $\theta_{\text{old}}$ and only train the new parameters $\theta_{\text{new}}$. The objective becomes:

$$\min_{\theta_{\text{new}}} \ \mathcal{L}_{\text{new}}(\theta_{\text{old}}^*, \theta_{\text{new}}) \tag{30}$$

This can be viewed as applying an infinitely strong regularization to keep $\theta_{\text{old}}$ unchanged:

$$\mathcal{L}_{\text{new}}(\theta) + \lambda \|\theta_{\text{old}} - \theta_{\text{old}}^*\|^2 \quad \text{with} \quad \lambda \to \infty \tag{31}$$

## A.2 PROOF OF THEOREM 4.1

*Proof.* Under the infinite-width NTK approximation, the parameter update from task $t - 1$ to task $t$ is given by Eq.equation 17:

$$\Delta\theta_t = -\left(J_t^\top J_t + \lambda\,\Omega_t\right)^{-1} J_t^\top e_t. \tag{32}$$

The change in function output for earlier task $s$ is:

$$\Delta f_s = f_{\theta_t}(X_s) - f_{\theta_{t-1}}(X_s) \tag{33}$$

$$= J_s \Delta\theta_t \tag{34}$$

$$= -J_s\left(J_t^\top J_t + \lambda\,\Omega_t\right)^{-1} J_t^\top e_t \tag{35}$$

$$= -K_{st}\left(K_{tt} + \lambda\,\widetilde{K}_t\right)^{-1} e_t, \tag{36}$$

where we have used the definitions $K_{st} = J_s J_t^\top$, $K_{tt} = J_t J_t^\top$, and $\widetilde{K}_t = J_t \Omega_t J_t^\top$.

To bound $\|\Delta f_s\|_2$, we apply the submultiplicativity of the spectral norm:

$$\|\Delta f_s\|_2 = \|K_{st}\left(K_{tt} + \lambda\,\widetilde{K}_t\right)^{-1} e_t\|_2 \tag{37}$$

$$\leq \|K_{st}\|_2 \cdot \|(K_{tt} + \lambda\,\widetilde{K}_t)^{-1}\|_2 \cdot \|e_t\|_2. \tag{38}$$

Since $K_{tt} = J_t J_t^\top$ is positive semidefinite and $\widetilde{K}_t = J_t \Omega_t J_t^\top$ is also positive semidefinite (assuming $\Omega_t \succeq 0$), the matrix $K_{tt} + \lambda\widetilde{K}_t$ is positive definite for $\lambda > 0$. Therefore:

$$\|(K_{tt} + \lambda\,\widetilde{K}_t)^{-1}\|_2 = \frac{1}{\sigma_{\min}(K_{tt} + \lambda\,\widetilde{K}_t)}. \tag{39}$$

Combining these results yields:

$$\|\Delta f_s\|_2 \leq \frac{\|K_{st}\|_2}{\sigma_{\min}(K_{tt} + \lambda\,\widetilde{K}_t)}\|e_t\|_2. \tag{40}$$

$\square$

## A.3 PROOF OF THEOREM 4.2

*Proof.* We start from the bound established in Theorem 4.2:

$$\|\Delta f_s\|_2 \leq \frac{\|K_{st}\|_2}{\sigma_{\min}(K_{tt} + \lambda\,\widetilde{K}_t)}\|e_t\|_2. \tag{41}$$

The key insight is to lower bound $\sigma_{\min}(K_{tt} + \lambda\,\widetilde{K}_t)$ using the spectral properties of $\Omega_t$ on the gradient-overlap subspace.

Let $\mathcal{S} = \mathrm{span}\{J_t^\top J_s : s < t\}$ denote the gradient-overlap subspace. By assumption, $\sigma_{\min}(\Omega_t) \geq \mu > 0$ on $\mathcal{S}$.

For any vector $v \in \mathbb{R}^{n_t}$, we can decompose it as $v = v_\parallel + v_\perp$, where $v_\parallel$ lies in the range space of $J_t^\top$ (which contains $\mathcal{S}$) and $v_\perp$ is orthogonal to it.

The quadratic form can be written as:

$$v^\top(K_{tt} + \lambda\widetilde{K}_t)v = v^\top J_t J_t^\top v + \lambda v^\top J_t \Omega_t J_t^\top v \tag{42}$$

$$= \|J_t^\top v\|_2^2 + \lambda(J_t^\top v)^\top \Omega_t(J_t^\top v). \tag{43}$$

Since $J_t^\top v \in \mathrm{range}(J_t^\top) \supseteq \mathcal{S}$ and $\sigma_{\min}(\Omega_t) \geq \mu$ on $\mathcal{S}$, we have:

$$(J_t^\top v)^\top \Omega_t(J_t^\top v) \geq \mu\|J_t^\top v\|_2^2. \tag{44}$$

Therefore:

$$v^\top (K_{tt} + \lambda \widetilde{K}_t) v \geq \|J_t^\top v\|_2^2 + \lambda \mu \|J_t^\top v\|_2^2 \tag{45}$$

$$= (1 + \lambda \mu) \|J_t^\top v\|_2^2. \tag{46}$$

For vectors $v$ in the range space of $K_{tt}$, we have $\|J_t^\top v\|_2^2 = v^\top K_{tt} v \geq \sigma_{\min}(K_{tt}) \|v\|_2^2$.

Thus, for such vectors:

$$v^\top (K_{tt} + \lambda \widetilde{K}_t) v \geq (1 + \lambda \mu) \sigma_{\min}(K_{tt}) \|v\|_2^2. \tag{47}$$

This implies:

$$\sigma_{\min}(K_{tt} + \lambda \widetilde{K}_t) \geq \sigma_{\min}(K_{tt})(1 + \lambda \mu). \tag{48}$$

Substituting this bound into the result from Theorem 1:

$$\|\Delta f_s\|_2 \leq \frac{\|K_{st}\|_2}{\sigma_{\min}(K_{tt})(1 + \lambda \mu)} \|e_t\|_2. \tag{49}$$

$\square$

### A.4 BACKWARD TRANSFER

We evaluated the backward transfer results (BWT) of each model in the Tab. 3

Table 3: Backward transfer of various methods with memory size 500

| Method | MNIST | | CIFAR-10 | | CIFAR-100 | |
|---|---|---|---|---|---|---|
| | Class-IL | Task-IL | Class-IL | Task-IL | Class-IL | Task-IL |
| fine-tuning | -96.39$_{\pm0.12}$ | -46.24$_{\pm2.12}$ | -89.68$_{\pm0.96}$ | -62.46$_{\pm0.78}$ | -78.94$_{\pm0.81}$ | -67.34$_{\pm0.79}$ |
| AGEM | -94.01$_{\pm1.16}$ | -14.26$_{\pm1.18}$ | -88.5$_{\pm1.56}$ | -45.43$_{\pm2.32}$ | -78.03$_{\pm0.78}$ | -59.28$_{\pm1.08}$ |
| GSS | -62.88$_{\pm2.67}$ | -7.73$_{\pm3.99}$ | -82.17$_{\pm4.16}$ | -33.98$_{\pm1.54}$ | —— | —— |
| HAL | -62.21$_{\pm4.34}$ | -5.41$_{\pm1.10}$ | -49.29$_{\pm2.82}$ | -13.60$_{\pm1.04}$ | —— | —— |
| ER | -45.35$_{\pm0.07}$ | -3.54$_{\pm0.35}$ | -74.87$_{\pm1.38}$ | -16.81$_{\pm0.97}$ | -75.24$_{\pm0.76}$ | -31.98$_{\pm1.35}$ |
| DER++ | -22.38$_{\pm4.41}$ | -4.66$_{\pm1.15}$ | -53.89$_{\pm1.85}$ | -14.72$_{\pm0.96}$ | -64.6$_{\pm0.56}$ | -27.21$_{\pm1.23}$ |
| +refresh | -22.03$_{\pm3.89}$ | -4.37$_{\pm1.25}$ | -53.51$_{\pm0.70}$ | -14.23$_{\pm0.75}$ | -63.90$_{\pm0.61}$ | -25.05$_{\pm1.05}$ |
| +K-RECON | -21.86$_{\pm1.21}$ | -4.34$_{\pm0.85}$ | -53.22$_{\pm0.76}$ | -14.35$_{\pm0.52}$ | -63.85$_{\pm1.02}$ | -24.98$_{\pm0.79}$ |

### A.5 COMPUTATION EFFICIENCY

We evaluated the computation efficiency of our model in the Tab. 4. As can be seen from the table, although the method proposed in this paper increases the computation time of the original method, it has a certain computational efficiency advantage compared to the method that requires calculating the Fisher information matrix.

Table 4: Computational efficiency of K-RECON on CIFAR-100 with one epoch training

| CIFAR-100 | DER++ | refresh | K-RECON |
|---|---|---|---|
| time(seconds) | 8.4 | 15.2 | 12.6 |

### A.6 LIMITATIONS

Although K-RECON enhances the plasticity of continual learning at the neuronal level by introducing a neuro-cognitive inspired reconsolidation mechanism, there are still some limitations. First, the proposed framework is mainly evaluated on image classification benchmarks with clearly defined task boundaries (e.g., CIFAR-10, CIFAR-100, Tiny-ImageNet). Its effectiveness in more complex environments—such as task-agnostic continual learning and cross-modal settings—would require additional mechanisms for task reasoning or dynamic memory routing to extend K-RECON to such challenging settings.

Second, while it is based on cognitive neuroscience, the selective reactivation and adaptive consolidation module abstractions at the conceptual level diverge from biological processes; deeper alignment with neural mechanisms (such as hierarchical memory systems, neuromodulatory signals, or time-dependent plasticity) could enhance the model's interpretability and biological verifiability.

Finally, the method introduces additional computational components, including the dynamic construction of a parameter importance matrix and a two-stage regularization process. While effective in the current experiments, this increased overhead may hinder deployment in resource-constrained environments or large-scale continual learning applications. In future work, our goal is to explore more lightweight approximations of memory consolidation and investigate adaptive mechanisms based on learning representations for long-term predictability.

