# OpenReview forum: "Toward Balanced Continual Learning via Fine-Grained Neuronal Intervention Inspired by Memory Consolidation"
_ICLR.cc/2026/Conference — ICLR 2026 Conference Desk Rejected Submission_

### Official Review · Reviewer_PwRy · 2025-10-27

**Soundness:** 2
**Presentation:** 1
**Contribution:** 3
**Rating:** 4
**Confidence:** 4

**Summary:**

This paper introduces a method termed K-RECON to address the stability-plasticity dilemma within continual learning (CL), drawing inspiration from the neurobiological mechanism of memory reconsolidation. The authors argue that existing CL methods rely on coarse-grained, network-level regularization. In contrast, K-RECON is designed to operate at a fine-grained, "neuron-level". The method is presented as a plug-and-play enhancement for existing CL frameworks and is supported by a theoretical analysis under a unified regularization framework. Experiments on CIFAR-10, CIFAR-100, and Tiny-ImageNet show that K-RECON consistently improves the performance of baseline methods.

**Strengths:**

+ The paper provides a novel insight by unifying several existing CL paradigms (e.g., regularization, replay, distillation) under a common quadratic regularization framework (Section 2). This effectively contextualizes the paper's contribution.

+ The method is evaluated across both Task-IL and Class-IL scenarios, effectively demonstrating K-RECON's utility as a plug-and-play module that enhances various established baselines.

+ The paper includes useful ablation studies and analyses to support its claims. For instance, Figure 2 well visualizes how the proposed method alleviates the problem of excessive regularization intensity compared to EWC.

**Weaknesses:**

+ The paper frequently uses the terms "neuron-level" and "neuronal clusters" to describe the proposed approach. However, the technical implementation described in Sections 2 and 3 defines the regularization matrix $\Omega$ over the parameters $\theta$, which is similar to existing "Parameter-Level" methods (e.g., EWC or SI). The connection between a specific parameter's importance value ($\Omega_i$) and a functional "neuronal cluster" is not explicitly defined or justified, creating a disconnect between the method's biological inspiration and its actual implementation.

+ The experimental baselines appear somewhat dated. Most of the chosen comparison methods (e.g., SI, OEWC, GEM, HAT, DER++) were proposed in or before 2021. This makes it difficult to assess the true performance of K-RECON against the current state-of-the-art. Including comparisons to more recent and stronger CL methods from the last 2-3 years would be essential to robustly validate the method's claimed advancements.

+ The paper's presentation could be significantly improved. For instance, the motivation in the Introduction is brief and could be strengthened. Moreover, Figure 1 is confusing and difficult to interpret, hindering the reader's understanding of the model's workflow.

**Questions:**

+ Could the authors provide a more detailed analysis of the computational and memory complexity of K-RECON? Specifically, how does the cost of storing an importance matrix $\Omega^{(k)}$ for *every* past task and then performing pairwise similarity calculations and Pareto optimization scale as the number of tasks ($t$) increases?

+ Could the authors clarify the mapping between the parameter-space importance matrix $\Omega$ and the "neuronal clusters" mentioned conceptually? Clarifying this would significantly strengthen the link between the biological inspiration and the technical implementation.

---

### Official Review · Reviewer_Ka9F · 2025-11-01

**Soundness:** 3
**Presentation:** 1
**Contribution:** 3
**Rating:** 2
**Confidence:** 3

**Summary:**

The paper proposes K-RECON, a plug-and-play continual learning method inspired by memory reconsolidation in biological learning for tackling the stability-plasticity tradeoff in continual learning. It adds two neuron-level mechanisms: a Selective Reactivation Module that aggregates task-wise parameter-importance vectors using cosine similarity and a Pareto objective, and an Adaptive Consolidation Module that applies attention-weighted protection/forgetting via thresholds to balance stability and plasticity. The authors place many continual learning methods (for addressing catastrophic forgetting) under a unified quadratic-regularization view and derive NTK-based closed-form updates and forgetting bounds. The authors empirically validate their method on class incremental and task incremental variants of CIFAR-10/100 and Tiny-ImageNet (buffer=500), demonstrating a consistent improvement over existing baseline methods.

**Strengths:**

- The paper provides a nice and succinct summary and unified view of existing regularization-based methods for addressing catastrophic forgetting section 2.
- The proposed method is well motivated by biological learning.
- The plug-and-play nature of K-RECON is a positive of this method.
- The empirical results demonstrate that the proposed K-RECON method consistently improves the performance of different continual learning methods while serving as a plug-and-play module.

**Weaknesses:**

- There are quite a few critical typos in the paper, for instance I believe that some $\omega$'s should be $\Omega$'s or $f_{\theta_{t-1}}$ should be $f_{\theta_t}$ is line 259. These should be addressed as they make parsing the mathematical statements for the reader difficult.
- Section 3 is difficult to follow and lacks some details. See my clarifying questions below. It would be useful to the reader to define different variables more formally, and how they are computed, selected, or updated: for instance: $\Omega^{(i)}$, $\lambda_k$, $f$.
- While the motivating example of biological learning is a positive, the repetition of these ideas when introducing the method is quite jargon-y and hand-wavey. For instance in Section 3, what exactly is Pareto optimization here? It would be useful to define what exactly is being referred to as Pareto optimization here.
- Section 4 also suffers from similar issues of clarity and communication as in section 2. For instance, the motivating analysis using NTK Linearization could be introduced better. For instance, what is $\Omega_t$ here, is it a general $\Omega$ used in section 2 or is the specific $\Omega$ derived in section 3? Furthermore, the presentation of K-Recon could be could greatly improved by presenting the algorithm in pseudo-code. Here lines 316 and 317 seem to include an argument, or proof, when introducing K-RECON, ideally for clarity the characterization of the method should be separate, as a theorem, from the description or pseudocode of the method.

**Questions:**

- In equation (11) what is $\omega_{\text{egg}}$. What exactly is being minimized in equation (11)?
- How is the adaptive weight $\lambda_k$ computed?
- How are $\Omega^{(i)}$ estimated, as introduced and stated in Section 3?
- On line 245, what if $f$ the attention application factor exactly? Is this a trainable parameter, if so how is it initialized?
- On line 259, what is $y_t$ and NC-ParetoCL.

---

### Official Review · Reviewer_PXDL · 2025-11-01

**Soundness:** 2
**Presentation:** 2
**Contribution:** 3
**Rating:** 4
**Confidence:** 4

**Summary:**

This paper proposes K-RECON, a neuron-level continuous learning framework inspired by memory reconsolidation. The method comprises two core modules: Selective Reactivation: based on the similarity of the inter-task importance matrix Ω, selectively reactivates old knowledge relevant to the current task; Adaptive Consolidation: through Pareto optimisation and thresholding operations, it integrates weighted importance information across different tasks to consolidate critical memories at the parameter level. Empirical results on CIFAR-10, CIFAR-100, and Tiny-ImageNet demonstrate that K-RECON achieves sustained performance improvements over multiple mainstream baseline methods, including EWC, SI, and DER++.

**Strengths:**

1. The paper proposes a framework named K-RECON, inspired by memory reconsolidation, which abstracts biological memory processes into two computable modules: Selective Reactivation and Adaptive Consolidation. This offers a novel perspective.

2. The structure of the entire text is clear, with consistent references to figures and equations.

3. The abstract and discussion section emphasise that the model achieves an ‘automatically tuned equilibrium between stability and plasticity’ across different tasks, while redefining the memory retention mechanism in continuous learning from the perspective of neuronal interactions.

**Weaknesses:**

1. Figure 1 fails to explain the significance of the blue blocks, dashed lines, and colour-coded zones, nor does it clarify their correspondence with the importance matrix Ω or task identifiers. I observe no indication of how incremental learning is implemented. The cortical images presented in the second phase lack biological grounding, appearing more ornamental than scientifically substantiated.

2. The appendix merely states adherence to the standard CL protocol without providing relevant network architecture, hyperparameters, data partitioning, or random seed information. It also fails to describe the data partitioning method, task sequence, random seed, or number of repetitions. Consequently, the experiments cannot be reproduced, casting doubt on the reliability of the results.

**Questions:**

1. Can the author provide complete pseudocode or an algorithmic flowchart, including the calculation and update details for Ω?

2. The introduction asserts that the method can ‘automatically adjust the trade-off between stability and plasticity’. Might the authors provide specific rules or algorithmic details?

3. Equation (11) updates the current task parameters $\Omega_i$ by aggregating historical task parameters $\Omega^{(k)}$ through the weighted sum $\Omega_i = \min_{\Omega_\text{agg}} \sum_{k=1}^t \lambda_k \|\Omega_\text{agg} - \Omega^{(k)}\|$, where $\lambda_k$ is described in the paper as an adaptive weight. However, it remains unclear how $\lambda_k$ is determined or controlled in practice.May the author explain how it is configured and updated?

---

### Note · Program_Chairs · 2026-01-17
**Submission Desk Rejected by Program Chairs**

The following references in this submission do not refer to real documents and/or have major errors in bibliographic information:

 Hyojin Kang, Taesik Kim, and Taesup Moon. Plasticity gradient for efficient continual learning. In Advances in Neural Information Processing Systems (NeurIPS), 2022.